# Alpha-1,4-transglycosylation Activity of GH57 Glycogen Branching Enzymes Is Higher in the Absence of a Flexible Loop with a Conserved Tyrosine Residue

**DOI:** 10.3390/polym15132777

**Published:** 2023-06-22

**Authors:** Hilda Hubertha Maria Bax, Marc Jos Elise Cornelis van der Maarel, Edita Jurak

**Affiliations:** Bioproduct Engineering, Engineering and Technology Institute Groningen, University of Groningen, Nijenborgh 4, 9747 AG Groningen, The Netherlands; h.h.m.bax@rug.nl (H.H.M.B.); m.j.e.c.van.der.maarel@rug.nl (M.J.E.C.v.d.M.)

**Keywords:** glycogen branching enzyme, alpha-1,4-transglycosylation, flexible loop

## Abstract

Starch-like polymers can be created through the use of enzymatic modification with glycogen branching enzymes (GBEs). GBEs are categorized in the glycoside hydrolase (GH) family 13 and 57. Both GH13 and GH57 GBEs exhibit branching and hydrolytic activity. While GH13 GBEs are also capable of α-1,4-transglycosylation, it is yet unknown whether GH57 share this capability. Among the four crystal structures of GH57 GBEs that have been solved, a flexible loop with a conserved tyrosine was identified to play a role in the branching activity. However, it remains unclear whether this flexible loop is also involved in α-1,4-transglycosylation activity. We hypothesize that GH57 GBEs with the flexible loop and tyrosine are also capable of α-1,4-transglycosylation, similar to GH13 GBEs. The aim of the present study was to characterize the activity of GH57 GBEs to investigate a possible α-1,4-transglycosylation activity. Three GH57 GBEs were selected, one from *Thermococcus kodakarensis* with the flexible loop and two beta-strands; one from *Thermotoga maritima*, missing the flexible loop and beta-strands; and one from *Meiothermus* sp., missing the flexible loop but with the two beta-strands. The analysis of chain length distribution over time of modified maltooctadecaose, revealed, for the first time, that all three GH57 GBEs can generate chains longer than the substrate itself, showing that α-1,4-transglycosylation activity is generally present in GH57 GBEs.

## 1. Introduction

Glycogen, the primary energy storage molecule in animals, fungi, and bacteria is a complex biomolecule consisting of anhydroglucopyranose units linked via α-1,4- and α-1,6-*O*-glycosidic bonds [1,2,3]. The glucose residues form a spherical polymer of 20–50 nm and with a high degree of branching due to α-1,6-linkages [3,4,5,6]. The chains have an average DP of 6–14 anhydroglucopyranose units [3,4,5,7,8] and branch points are distributed in a random pattern [9]. The glycogen particle size, density, and fine structure are key to its function as energy storage unit, leading to more or less bioavailability, depending on the organism’s energy requirements [10,11]. Bacteria growing in relatively adverse environments produce glycogen with shorter branched chain length compared to bacteria living in nutrient-rich environments. Glycogen consisting of shorter chains is broken down more slowly, therefore enabling survival over longer periods of time [11].

The glycogen particles are synthesized by the interplay of several enzymes, including ADP-glucose pyrophosphorylase, glycogen synthase and glycogen branching enzymes (GBEs). These enzymes all work in concert in the classical GlgC–GlgA pathway to build the glycogen particle and define its final structure [3,12,13]. Additionally, the other pathway in bacteria to synthesize glycogen is the GlgE pathway, in which maltosyl transferase (GlgE) generates maltose and extends it to maltotetraose and increases its size until it is long enough to be used by GBEs [3,5]. In both pathways, branch points (α-1,6-linkages) are introduced by GBEs. These widely distributed enzymes introduce branches by transferring a chain segment cleaved off from one chain onto a new chain via the formation of an α-1,6-linkage. GBEs are essential for creating the glycogen structure and play a key role in defining the final structure of the glycogen particles.

Branching enzymes or 1,4-α-glucan: 1,4-α-glucan-6-glucosyltranferases (EC 2.4.1.18) are categorized, based on their primary amino acid sequence and conserved motifs, in either the glycoside hydrolase (GH) family 13 or 57 [14,15,16]. Within the family GH13, GBEs are categorized in the subfamilies GH13_8 (both bacterial and eukaryotic GBEs) and GH13_9 (almost exclusively bacterial GBEs) [15,17,18,19,20].

Both GH13 and GH57 GBEs consist of three domains, with domain A of GH13 GBEs exhibiting a (β/α)_8_-barrel fold, whereas the GH57 GBE domain A has a (β/α)_7_-barrel fold [9,13,14,21,22,23]. GH13 GBEs have three while GH57 GBEs have two catalytic site residues. The catalytic nucleophile and proton donor in GH13 GBEs are, respectively, an aspartic acid and a glutamic acid, whereas in GH57 GBEs these are a glutamic acid and an aspartic acid [15,21]. The third catalytic site residue in GH13 GBEs, the transition state stabilizer, is a second aspartic acid [24]. GH57 GBEs have a histidine, which performs the function of the stabilizer [23].

While the catalytic mechanism of GH13 GBEs has been described in detail, the mechanism of GH57 GBEs is poorly understood. In the double displacement mechanism, as schematically illustrated by Hayashi et al. (2017) and Gaenssle et al. (2021) [9,25], the first step is the entry of a donor glycoside chain into the catalytic site, followed by the cleaving of an α-1,4-linkage. Part of the donor chain is released, and a covalent enzyme-substrate intermediate is formed with the remaining segment. In the second step, a new glycoside acceptor chain binds to the enzyme and the retained segment from the donor chain is transferred to the acceptor chain, forming an α-1,6-linkage and creating a new branch [9,26,27]. In addition to this branching activity, both GH13 and GH57 GBEs exhibit hydrolytic activity. The hydrolytic activity appears to be more common for GH57 compared to GH13 GBEs, as GH57 GBEs have a higher hydrolytic activity relative to branching activity. This is the result of the transfer of the remaining donor fragment onto a water molecule instead of a new acceptor chain, resulting in the release of a free chain [5,21,23,28,29]. Next to branching and hydrolytic activity, GH13 GBEs are also capable of α-1,4-transglycosylation, resulting in elongated products [30,31]. The elongated products can be used as a donor or acceptor substrate again [25,31].

The family GH57 GBEs are thought to have a catalytic mechanism very similar to that of GH13 GBEs, although they lack one of the three catalytic site residues of GH13 GBEs [21]. In contrast to GH13 GBEs, the mechanism and role of GH57 GBEs are less clear. Currently, only four crystal structures of GH57 GBEs have been solved [21,22,23,32]. Of these structures, three (*Pyrococcus horikoshii, Thermococcus kodakarensis,* and *Thermus thermophilus*) displayed the presence of a flexible loop consisting of 19 amino acids with a highly conserved tyrosine at the tip [21,22,23]. This flexible loop and the tyrosine play a key role in the branching activity, as the mutations of the tyrosine or shortening of the loop have resulted in loss of activity [21,22]. A bioinformatic analysis showed that the flexible loop is absent in more than ninety percent of GH57 GBEs analyzed [33]. Additionally, two adjacent beta strands were identified as playing a role in the branching activity, alongside the flexible loop with a conserved tyrosine. Based on the presence or absence of the flexible loop, the tyrosine residue, and the beta-strands, GH57 GBEs can be divided into three groups. Only a small group, comprising 5.3% of the 2497 analyzed sequences, have both the flexible loop of 17–20 amino acids as well as the two beta strands [33]. Biochemical analysis confirmed that these GH57 GBEs can be defined as true GH57 GBEs due to their relatively high branching activity or α-1,6-transglycosylation activity [33]. It has been hypothesized that these true GH57 GBEs are also capable of α-1,4-transglycosylation, resulting in the production of elongated products, similar GH13 GBEs. However, to date, α-1,4-transglycosylation activity has not been reported for GH57 GBEs.

Therefore, the aim of the present study was to characterize the activity of GH57 GBEs using a highly defined substrate to investigate possible α-1,4-transglycosylation activity and gain a better understanding of the reaction mechanism of GH57 GBEs. Three GH57 GBEs were selected based on the presence or absence of the two beta strands and the flexible loop with the conserved tyrosine at the tip: *Thermococcus kodakarensis* (TkGBE57)*, Thermotoga maritima* (TmGBE57), and *Meiothermus* sp. (MsGBE57). TkGBE57 possesses both the two beta strands and the flexible loop with a tyrosine residue and is referred to as the true glycogen branching enzyme [33]. TmGBE57 lacks both the flexible loop and the two beta strands, whereas MsGBE57 has the flexible loop with a tyrosine but lacks the beta strands [33].

These three GH57 GBEs, along with the GH13 GBE from *E. coli*, were used to elucidate the mode of action and changes in chain length distribution on the highly defined substrate maltooctadecaose (MD18). The GH57 GBEs were able to generate longer linear chains, although to a lesser extent than GH13 GBEs, showing that α-1,4-transglycosylation activity is not restricted to GH13_9 GBEs but is also present in the majority of GH57 GBEs.

## 2. Materials and Methods

### 2.1. Materials

Linear maltodextrin MD18 (Maltooctadecaose) was purchased from CarboExpert (Yuseong-gu, Republic of Korea). Isoamylase was obtained from *Pseudomonas* sp. (E-ISAMY, 200 U/mL) and pullulanase M1 from *Klebsiella planticola* (E-PULKP, 650 U/mL) were obtained from Megazyme (Bray, Ireland). Glycogen branching enzymes of *Escherichia coli* (EcGBE13) was purchased from Creative Enzymes (Shirley, NY, USA), and MagicMedia and HisPur^TM^ Ni-NTA Resin were purchased from ThermoFischer Scientific (Waltham, MA, USA). All chemicals were of analytical grade or higher.

### 2.2. Enzyme Production and Purification

The genes encoding the GBE from *Thermococcus kodakarensis* KOD1 (TkGBE57, WP_011250387.1)*, Thermotoga maritima* SMB8 (TmGBE57, WP_004081707.1), and *Meiothermus* sp. PNK-Is4 (MsGBE57, WP_129865543.1) were codon-optimized by GenScript, cloned into a pRSET-B (TkGBE57 and TmGBE57) and pET28a(+) (MsGBE57) vector containing a C-terminal His-Tag for purification (GenScript USA Inc., Piscataway, NY, USA), and expressed in *E. coli* BL21(DE3). Each strain was grown on ampicillin or kanamycin-containing agar plates, from which single colonies were selected for starter cultures. The starter cultures were grown overnight in LB media (Luria–Bertani media, 1% NaCl, 1% tryptone, 0.5% yeast extract) supplemented with 50 µg/mL ampicillin (TkGBE57) or kanamycin (TmGBE57 and MsGBE57) at 37 °C and 150 rpm. From the starter cultures, 25 mL was transferred to the main 500 mL culture of MagicMedia^TM^, supplemented with 50 µg/mL ampicillin or kanamycin and grown for 24 h at 30 °C and 150 rpm. All cells were harvested via centrifugation (5000× *g*, 10 min, 4 °C), resuspended in lysis buffer (20 mM sodium phosphate, pH 7.4, 500 mM NaCl, 20 mM Imidazole), and lysed via sonication on ice (for 10 min with cycles of 30 s on and 30 s off; amplitude 20%, pulse 50%). The cell debris were then collected through centrifugation (12,000× *g*, 15 min, 4 °C). All cell extracts were subsequently purified using an ÄKTA system (GE Healthcare, Chicago, IL, USA) with a 5 mL HisPur^TM^ Ni-NTA column. The column was equilibrated with 20 mM sodium phosphate buffer (pH 7.4) containing 500 mM NaCl and 20 mM Imidazole with a flow rate of 1.0 mL/min. The flow rate was maintained for all following steps. Protein was loaded onto the column and washed until the UV signal (280 nm) stabilized. Bound protein was eluted with eluents A (20 mM sodium phosphate, pH 7.4, 500 mM NaCl, 20 mM Imidazole) and B (20 mM sodium phosphate, pH 7.4, 500 mM NaCl, 500 mM Imidazole) with the following gradient profile: 0–10 min (0–50% B), 10–20 min (50% B), 20–30 min (50–100% B), and 30–40 min (100% B). The fractions containing the desired proteins were collected, desalted via buffer exchange (20 mM sodium phosphate, pH 7.4), and concentrated using an Amicon^®^ Ultra filter (30,000 MWCO, 15 mL). The purified proteins were stored at −80 °C. Protein concentrations were measured with Pierce BCA Protein Assay (Thermo Fisher Scientific Inc., Waltham MA, USA) and the purity of the proteins was determined by SDS-PAGE (Appendix A).

### 2.3. Enzyme Activity with Iodine Assay

The iodine assay was used to test the activity on potato amylose V (AVEBE, The Netherlands) [19]. GBEs were incubated in different concentrations (30–300 µg/mL) with 1 mg/mL potato amylose in 50 mM sodium phosphate buffer, pH 7.5 at 50 °C. At every minute for 10 min, 15 µL aliquots were taken from the enzyme reactions and mixed with 100 µL freshly prepared iodine reagent (0.26% KI, 0.026% I^2^, 5 mM HCl). After the transfer of the last aliquot, the absorbance of the iodine–amylose complex was measured at 610 nm using a spectrophotometer (SpectraMax Plus 384 Microplate Reader, Molecular Devices, Sunnyvale, CA, USA). One unit of activity is defined as the decrease in absorbance of 1.0 per min at 610 nm (Appendix A).

### 2.4. Enzyme Reactions and Analysis with Reducing End Assay

The branching and non-branching activity of the GBEs was calculated by first incubating the enzymes with 2.5 mg/mL potato amylose in 10 mM sodium phosphate buffer slowly rotating head-over-tail at 50 °C, with an enzyme dose of 4 U/g substrate (units based on the iodine assay). The non-branching activity of TkGBE57 was analyzed with an enzyme dose of 20 U/g substrate as the activity at 4 U/g substrate was too low to detect. Samples were incubated for 0.5, 1, 1.5, 2, and 2.5 h and the reaction was stopped by boiling for 5 min. For the debranching reaction, the GBE-modified glucans were diluted twice in sodium acetate buffer, pH 4.5, and treated with isoamylase (1 U/mg substrate) and pullulanase (0.7 U/mg substrate) for 24 h at 40 °C.

All branched and debranched samples were analyzed with the pAHBAH reducing end assay by mixing 50 μL sample at a concentration of 2.5 mg/mL and 1.0 mg/mL amylose for branched and debranched samples, respectively (200 μL pAHBAH solution containing 1/5 of 5% 4-hydroxybenzoic acid hydrazide in 0.5 M HCl and 4/5 of 0.5 M NaOH; 30 min at 70 °C). The absorbance was measured at 490 nm using a spectrophotometer. Samples were measured in triplicates and D-glucose (Sigma Aldrich, St. Louis, MI, USA) was used as a standard. The linear chains were determined from the reducing ends of the branched product and the branched chains from the reducing ends of the debranched product minus the linear chains. One unit of activity is defined as 1 µmol reducing ends released or transferred per minute (Appendix A).

Due to the previously described branching, hydrolytic, and elongation activity of GBEs, here we introduce a specific definition of activity as U^B^ (unit-branching activity), which represents only the branching activity without the side-activities or non-branching activities, thus allowing a fair comparison of the activity and mode of action of GBEs. The increase in reducing ends from branched to debranched products represents the branching activity, U^B^, defined as:$$U^{B}=\frac{\Delta RE\;after\;debranching\;\left[\mu mol\right]-\Delta RE\;before\;debranching\left[\mu mol\right]}{\Delta time\;\left[min\right]}$$

One $U^{B}$ of activity is defined as 1 μmol of reducing ends transferred to a branching point per minute.

The branching and debranching reaction was repeated with MD18 as substrate (0.5 mg/mL), with an enzyme dose of 0.5 U^B^/g substrate for TkGBE57, TmGBE57, and MsGBE57 (units, U^B^, based on branching activity on amylose). For TkGBE57, a higher enzyme dose (1 U^B^/g substrate) was needed to detect the non-branching activity, whereas for EcGBE13, a lower enzyme dose (0.125 U^B^/g substrate) was needed to detect a linear increase in reducing ends.

### 2.5. Chain Length Distribution with Anion Exchange Chromatography

To investigate and compare the mode of action of the GBEs an incubation was carried out with MD18 (0.5 mg/mL) and an enzyme dose of 1 U^B^/g substrate and 50 mU^B^/g substrate for all enzymes (units, U^B^, based on branching activity on MD18). Both branched and debranched samples of this incubation were analyzed via high-performance anion exchange chromatography coupled with pulsed amperometric-EDet1 detection (HPAEC-PAD) with the gold standard PAD waveform, using a Dionex ICS-6000 system (ThermoFischer Scientific; Waltham, MA, USA) with a CarboPac^TM^ PA100 column. Samples were prepared by centrifugation (10,000× *g*, 10 min) and the dilution of the supernatant to 0.1 mg/mL maltodextrins. The injection volume was 10 µL and the flow rate was 0.25 mL/min. The samples were eluted using the eluents A (0.1 M NaOH) and B (0.1 M NaOH, 1 M NaOAc) with the following gradient profile: 0–50 min (5–40% B), 50–65 min (40–100% B), and 65–70 min (100% B), followed by a re-equilibration at 5% B. Chromatograms show representative samples and peak areas were calculated using the Chromeleon software version 7.2.9. The increase in linear chains was calculated as Δ peak area, representing the difference in peak area from untreated substrate to 24 h modification, and the Δ peak area of branched chains was calculated by the increase in peak area after the debranching of 24 h modified samples minus the increase in peak area after the debranching of untreated substrate.

### 2.6. Statistical Analysis

Data are presented as means of triplicates with standard deviations. The statistical analysis was conducted in Stata17 (StataCorp, College Station, TX, USA) using one-way analysis of variance (ANOVA) with a post hoc Bonferroni test and a 95% confidence interval.

## 3. Results and Discussion

### 3.1. Definition of Activity for Glycogen Branching Enzymes

Three GH57 GBEs were selected based on the presence or absence of the beta strands and the flexible loop with conserved tyrosine at the tip: *Thermococcus kodakarensis* (TkGBE57), *Thermotoga maritima* (TmGBE57) and *Meiothermus* sp. (MsGBE57). Additionally, the GH13 GBE from *Escherichia coli* (EcGBE13) was included as a reference. As previously described [33], TkGBE57 possesses both the two beta strands and the flexible loop with a tyrosine, demonstrating a relatively high branching activity on amylose (480.0 mU/mg) compared to other GH57 GBEs [33]. The activity of these true glycogen branching enzymes is significantly lower than that of GH13 GBEs. TmGBE57 lacks both the flexible loop and the beta strands and has a reported branching activity of only 10.5 mU/mg on amylose. On the other hand, the third GH57 GBE, MsGBE57, has only the flexible loop with a tyrosine and exhibits a low branching activity of 29.1 mU/mg on amylose [33]. The enzymatic activity of selected and purified GBEs was verified on amylose using two different assays: the iodine assay and the reducing end assay. The iodine assay is a relatively fast and easy method for detecting the activity of glycogen branching enzymes, based on the decrease in long linear chains [25,34,35,36,37]. However, the iodine assay does not detect the alpha-1,4-transferase activity of GH13 GBEs, which results in the creation of longer chains [25]. On the other hand, the reducing end assay analyzes the increase in the reducing ends (RE) of substrate treated with GBEs over time, corresponding to the non-branching activity. The products of the incubation with GBEs are subsequently treated with debranching enzymes isoamylase and pullulanase M1, resulting in a further increase in reducing ends. The change in reducing end concentration after debranching is related to the branching activity of the GBEs. While the reducing end assay is useful for distinguishing between non-branching and branching activity, it does not give information about the alpha-1,4-transferase activity either. The branching activity detected with a reducing end assay is defined as U^B^ (unit-branching). To compare the branching or alpha-1,6 transferase activity of GBEs, a new definition of activity, U^B^, is introduced. One U^B^ of activity is defined as 1 μmol of reducing ends transferred to a branching point per minute.

### 3.2. Activity of GH57 Glycogen Branching Enzymes on Linear Maltodextrin

A highly defined linear maltodextrin consisting of 17 and 18 anhydro glucopyranose residues (MD18) was used as substrate to investigate the mode of action of TkGBE57, TmGBE57, MsGBE57, and EcGBE13 by analyzing the reducing ends and chain length distribution profile before and after debranching (Table 1)**.** The activity of TmGBE57 and MsGBE57 does not differ significantly; both enzymes have a branching to non-branching activity ratio (N:NB) of around 1 (1.2 ± 0.2 and 1.1 ± 0.0, respectively). In contrast, TkGBE57 has a significantly higher B:NB ratio (8.1 ± 2.3). Furthermore, TkGBE57 exhibits the highest total activity (105.4 ± 15.1 mU/mg E) among the three GH57 GBEs. For all GBEs used, the activity on MD18 is lower compared to amylose. In a study of Xiang et al., the branching activity on amylose for TmGBE57, TkGBE57, and MsGBE57 was determined to be 10.5 ± 0.3, 550.0 ± 5.5, and 29.1 ± 3.5 mU/mg, respectively [33]. TkGBE57 exhibits the highest activity on both amylose and MD18, which is hypothesized to be related to the presence of a flexible loop of 19 amino acids with a tyrosine at the tip and two beta-strands in the structure of the enzyme [33]. Interestingly, MsGBE57 showed higher activity on amylose compared to TmGBE57, although there were no significant differences between these two enzymes when incubated with MD18. Possibly, the higher activity of MsGBE57 on amylose is associated with the presence of the flexible loop with a tyrosine [33]. TmGBE57 and MsGBE57 exhibited similar B:NB ratios on amylose (5.7 ± 0.1 and 4.7 ± 1.7, respectively). This ratio is five times higher compared to the B:NB ratio on MD18; however, for both substrates, there were no significant differences between TmGBE57 and MsGBE57.

Compared to all three GH57 GBEs, both the branching and non-branching activity of EcGBE13 on MD18 is significantly higher. This corresponds with previous research indicating a higher activity for most GH13 GBEs compared to GH57 GBEs [29,36,37,38].

All three GH57 GBEs were incubated with MD18 based on the branching activity U^B^, using an equal enzyme dose of 1 U^B^ per gram of substrate. The increase in reducing ends was monitored over time, both before and after debranching (Figure 1). The total increase in reducing ends after incubation with the GBEs for 6 h and subsequent debranching was not significantly different for all three enzymes (958 ± 46, 927 ± 79, and 844 ± 71 μmol/g substrate, for TkGBE57, TmGBE57 and MsGBE57, respectively). TkGBE57 reached the maximum increase in reducing ends due to branching activity after only 2 h. MsGBE57 exhibited a linear increase over the 0 to 6 h time period, indicating lower branching activity compared to TkGBE57. The increase in reducing ends after 6 h of incubation with GBEs (without debranching) was similar for TmGBE57 and MsGBE57 (247 ± 34 and 258 ± 24 μmol/g S, respectively), whereas it was significantly lower for TkGBE57 (115 ± 20 μmol/g S). This is consistent with the B:NB ratio described earlier. Therefore, the reducing end profile in Figure 1 supports the high branching activity and B:NB ratio of TkGBE57, as shown in Table 1.

### 3.3. Chain Length Distribution of Modified Maltodextrin

The chain length distribution MD18 samples treated with GBEs for 24 h was analyzed before and after debranching (Appendix A). The presence of chains in the branched products indicated the presence of free linear chains. The chains in debranched products represented both branches and linear chains (Figure 2). The untreated MD18 substrate primarily consisted of linear chains with DP17 and 18. Upon the incubation of MD18 with three different GH57 GBEs, there was a significant increase in chains shorter than DP18, both before and after debranching. TkGBE57 predominantly generated linear chains of DP 6-8. TmGBE57 and MsGBE57 also produced minor amounts of linear chains in the range of DP 10-15. Previous studies on the modification of a maltodextrin mixture DP 10-40 with TkGBE57 and TtGBE57 (*T. thermophilus)* also demonstrated a preference for the linear chain lengths of DP 6-7 [38].

When compared to the branch lengths of MD18 modified with GH13 GBEs, GH57 GBEs exhibit less specificity in their branch length preferences. A study by Gaenssle et al. (2021) revealed that the GH13 GBEs from *Rhodothermus marinus*, *Petrotoga mobilis*, and *Butyrivibrio fibrisolvens* showed a preference for branch lengths of DP 9, DP 4-12, and DP 8-12, respectively [25]. Similarly, when modifying a maltodextrin mixture of DP 10-40 and several starches (waxy corn, tulip, potato, and wrinkled pea starch), GH13 GBEs exhibited a preference for branch lengths of DP 3-9, while GH57 GBEs preferred DP > 8. Regardless of the substrate, GH57 GBEs generally produce longer branches compared to GH13 GBEs [38].

The increase in peak heights from branched to debranched samples was highest for TkGBE57, indicating the highest branching activity (Figure 2). This observation aligns with the earlier analysis of increasing reducing ends. After 24 h, only 5.5%, 1.3%, and 0.5% (TkGBE57, TmGBE57, and MsGBE57, respectively) of the initial amount of DP17 and 18 chains remained. As shown in Figure 2, all three enzymes primarily generated linear chains, with DP7 being the predominant length. This chain length is relatively shorter compared to the average chain length of DP 6-14 typically found in glycogen. The presence of shorter chains leads to a denser glycogen structure, which is metabolized more slowly, providing a sustained energy supply over an extended period of time. This characteristic can be advantageous for organisms inhabiting harsh environments. *Thermococcus kodakarensis*, *Thermotoga maritima*, and *Meiothermus sp*. were isolated from geothermally heated marine surfaces or hot springs [33].

### 3.4. Alpha-1,4-transferase Activity of GH57 Glycogen Branching Enzymes

Previous research conducted by Gaenssle et al. [25] demonstrated the α-1,4-transglycosylation activity for GH13_9 GBEs. In order to investigate whether GH57 GBEs also possess this activity, the highly defined linear substrate MD18 was utilized, and chain length distribution analysis was employed. Figure 3 illustrates the increase in peak area for linear chains and branches larger than the initial substrate (>DP 18). The substrate was modified with TkGBE57, TmGBE57, MsGBE57, and EcGBE13 at two different concentrations (1 U^B^/g and 50 mU^B^/g). All three GH57 GBEs exhibited an augmentation in the quantity of chains longer than the initial DP18 substrate, demonstrating α-1,4-transglycosylation activity. Although this activity was notably lower compared to GH13 GBEs, it was not limited to the previously studied GH13_9 GBEs but was also present in GH57 GBEs.

At an enzyme dose of 1 U^B^/g, there is no significant difference in the quantities of longer chains between TkGBE57, TmGBE57, and MsGBE57. However, it is interesting to note that the amount of long linear chains increased significantly (Δ19.6 nC*min) for MD18 modified with TmGBE57 at an enzyme dose of 50 mU^B^/g. This higher elongation activity could be attributed to the availability of substrate. When there is a greater amount of acceptor available at a lower enzyme dose, the likelihood of creating an α-1,4-linkage is increased.

Figure 4 illustrates the distribution of chains > DP18 for TkGBE57, TmGBE57, and MsGBE57. It is important to highlight that the peak heights of TmGBE57 are significantly higher compared to the other two GH57 GBEs. Both TkGBE57 and MsGBE57 predominantly generate longer linear chains, primarily ranging from DP 28 to 30. These enzymes also produce branched elongated chains in the range of DP 21 to 28. In contrast, TmGBE57 yields a larger number of elongated chains, which are predominantly linear in nature. The elevated elongation activity exhibited by TmGBE57 may be associated with the structural elements of GH57 GBEs, as described by Xiang et al. [33]. TmGBE57 belongs to the largest group of GH57 GBEs (92%), which lacks the flexible loop and beta-strands associated with branching activity. In GBEs where the flexible loop is present, it occupies a position covering the catalytic cleft [33]. The absence of the flexible loop in TmGBE57 might contribute to its high α-1,4-transglycosylation activity, allowing the active site of the enzyme to be more accessible for acceptor substrate binding in a manner conducive to transfer to a new α-1,4-linkage.

In the incubation with a high enzyme dose (1 U^B^/g), the highest amounts of chains > DP19 were found with shorter incubation times (Figure 5). The elongated chains after modification with TkGBE57 continued increasing over 24 h. Modification with TmGBE57 and MsGBE57 showed an increase until 2 h and, thereafter, a decrease in elongated chains. The enzymes with a low branching over non-branching activity ratio (TmGBE57 and MsGBE57; Table 1) first create longer chains before using the elongated chains as new substrate. TkGBE57 has a higher branching over non-branching activity ratio, which resulted in lower amounts of elongated chains in the first 6 h of incubation with MD18. Only after 24 h, the amounts of elongated chains increased. A study by Tran et al. (2022) showed the ability of GH13 GBEs to produce elongated chains, and when long substrates are available, transglycosylation is gradually changed into branching activity [31]. Figure 5 also shows this concept of elongated products used again as substrates by GH57 GBEs, which was shown previously by Gaenssle et al. for GH13_9 GBEs [25].

## 4. Conclusions

The analysis of chain length distribution of modified MD18 over time, revealed for the first time that all three GH57 GBEs (TkGBE57, TmGBE57, and MsGBE57) are able to generate chains longer than the substrate. However, only to a lesser extent compared to GH13 GBEs. The results show that α-1,4-transglycosylation activity is not restricted to GH13_9 GBEs but is also present in the majority of GH57 GBEs [25].

The new definition of activity, U^B^, allowed the fair comparison of activities of GBEs. TmGBE57 on a low enzyme dose has a significantly higher α-1,4-transglycosylation activity compared to TkGBE57 and MsGBE57. A lower enzyme dose is related to a higher availability of acceptor substrate, which might explain the increase in elongated chains at a lower enzyme dose.

TmGBE57 showed the highest elongation activity, possibly related to the enzyme’s structural characteristics. In TmGBE57, as well as in 92% of all GH57 GBEs [33], a flexible loop with a tyrosine at the tip and the beta-strands are absent. The absence of the flexible loop could result in a more easily accessible catalytic side and a higher chance for the acceptor substrate to bind to the catalytic side in an orientation where α-1,4-transglycosylation activity is executed.

## Figures and Tables

**Figure 1 polymers-15-02777-f001:**
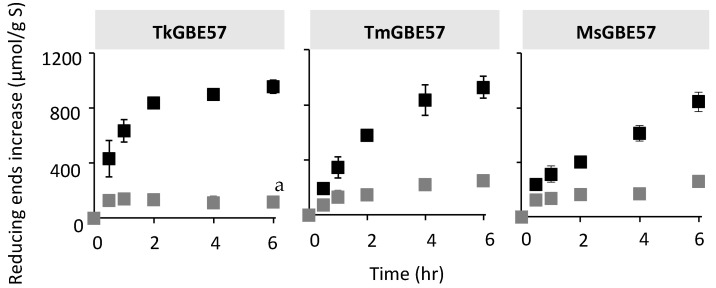
Increase in reducing ends of TmGBE57 (*Thermotoga maritima*), TkGBE57 (*Thermococcus kodakarensis*), or MsGBE57 (*Meiothermus* sp.) modified MD18 (1 U^B^/g S) before (grey) and after (black) debranching. ^a^ Significantly different from other enzymes after an incubation time of 6 h (*p* ≤ 0.05).

**Figure 2 polymers-15-02777-f002:**
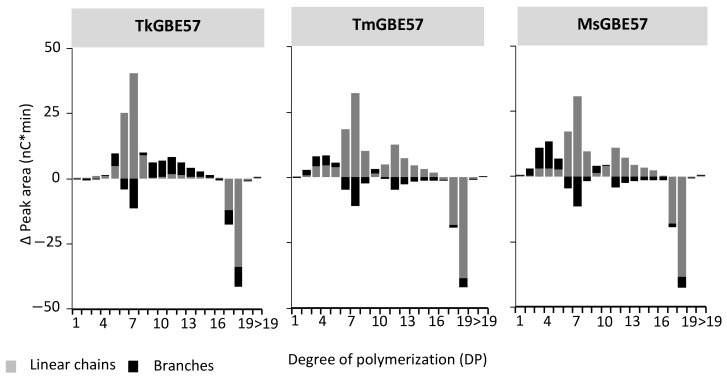
Estimated changes in peak areas of MD18 in linear and branched chains after treatment with TkGBE57 (*Thermococcus kodakarensis*), TmGBE57 (*Thermotoga maritima*), or MsGBE57 (*Meiothermus* sp.) at 1 U^B^/g S for 24 h compared to the untreated substrate. The Δ peak area of linear chains is calculated through the difference in peak area from untreated substrate to the 24 h modification of each DP, and the Δ peak area of branched chains is calculated via the increase in peak area after the debranching of 24 h modified samples minus the increase after the debranching of untreated substrate.

**Figure 3 polymers-15-02777-f003:**
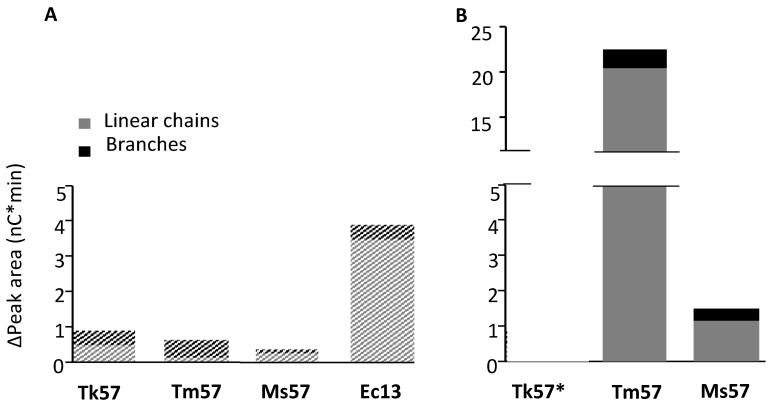
Estimated change in peak area of chains longer than MD18 in linear and branched chains after treatment with TkGBE57 (*Thermococcus kodakarensis*), TmGBE57 (*Thermotoga maritima*), MsGBE57 (*Meiothermus* sp.), or EcGBE13 for 24 h at two enzyme doses, namely (**A**) 1 U^B^/g S or (**B**) 50 mU^B^/g S, compared to the untreated substrate. The Δ peak area of linear chains is calculated by the difference in peak area from untreated substrate to 24 h modification of each DP, and the Δ peak area of branched chains is calculated by the increase in peak area after the debranching of 24 h modified samples minus the increase after the debranching of untreated substrate. * No detectable activity.

**Figure 4 polymers-15-02777-f004:**
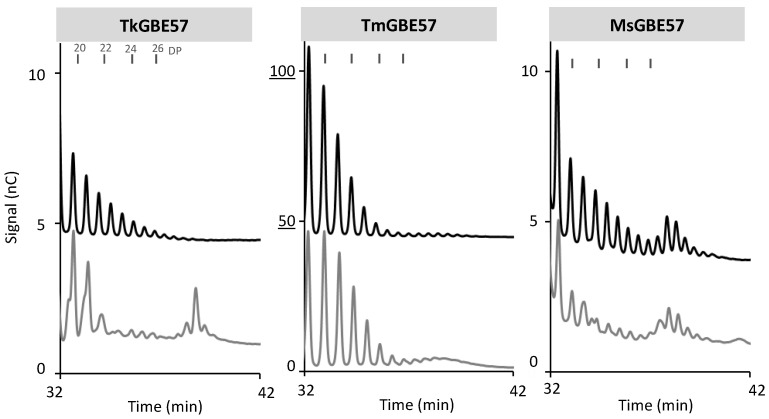
Chain length distribution of MD18 with enzyme TkGBE57 (*Thermococcus kodakarensis*), TmGBE57 (*Thermotoga maritima*), or MsGBE57 (*Meiothermus* sp.) at 50 mU^B^/g S for 24 h before (grey) and after (black) debranching compared to the untreated substrate.

**Figure 5 polymers-15-02777-f005:**
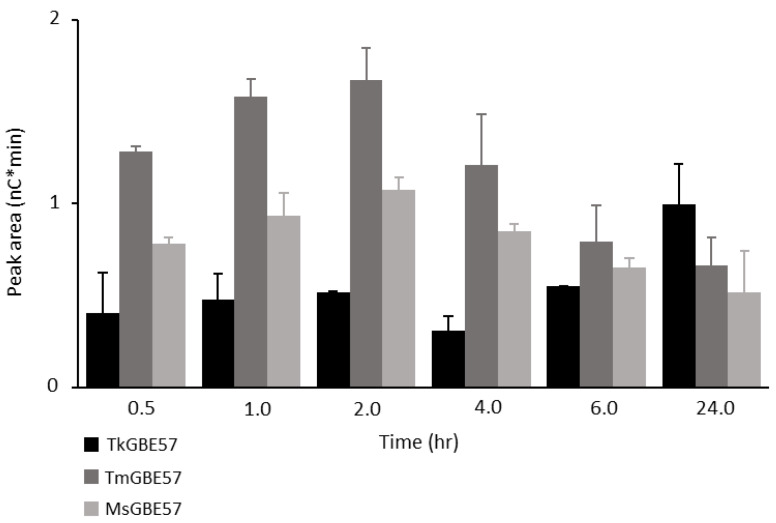
Sum of peak areas of linear and branched chains > DP19 from MD18 modified with TkGBE57 (*Thermococcus kodakarensis;* black), TmGBE57 (*Thermotoga maritima;* dark grey), or MsGBE57 (*Meiothermus* sp.; light grey) at 1 U^B^/g S for 24 h.

**Table 1 polymers-15-02777-t001:** Activity of TmGBE57(*Thermotoga maritima*), TkGBE57 (*Thermococcus kodakarensis*), MsGBE57 (*Meiothermus* sp.), and EcGBE13 (*Escherichia coli*) on MD18, analyzed as an increase in (non-branching activity) or transfer of (branching activity) reducing ends. Average of three independent measurements with standard deviation.

	TmGBE57	TkGBE57	MsGBE57	EcGBE13
**Non-branching activity [mU^NB^/mg E]**	4.4 ± 1.0	13.0 ± 5.6	5.2 ± 0.3	120.1 ± 0.9 ^a^
**Branching activity [mU^B^/mg E]**	5.0 ± 0.6	92.4 ± 9.5 ^a^	5.6 ± 0.2	286.9 ± 42.5 ^b^
**Ratio B:NB ***	1.2 ± 0.2	8.1 ± 2.3 ^a^	1.1 ± 0.0	2.4 ± 0.3

* ratio branching activity to non-branching activity. ^a,b^ Significantly different to the other enzymes (*p* ≤ 0.05)

## Data Availability

Raw data were generated at the University of Groningen, Faculty of Science and Engineering. Derived data supporting the findings of this study are available from the corresponding author [E. Jurak] on request.

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
