# Peer review of "Alpha-1,4-transglycosylation Activity of GH57 Glycogen Branching Enzymes Is Higher in the Absence of a Flexible Loop with a Conserved Tyrosine Residue"

_polymers, 2023, doi:10.3390/polym15132777_

Round 1

Reviewer 1 Report

The manuscript Alpha-1,4-transglycosylation activity of GH57 glycogen branching enzymes is higher in the absence of a flexible loop with a  conserved tyrosine residue. is a worthy work, the author has adopt state of the art methods for the preparation of target. The analysis of chain length distribution over time of modified MD18, revealed for 394 the first time, that all three GH57 GBEs (TkGBE57, TmGBE57 and MsGBE57) could gen- 395 erate chains longer than the substrate, although to lesser extent than GH13 GBEs.

Three GH57 GBEs were selected based on the presence or absence of the beta strands 224 and the flexible loop with conserved tyrosine at the tip; Thermococcus kodakarensis 225 (TkGBE57), Thermotoga maritima (TmGBE57) and Meiothermus sp. (MsGBE57). In addition, 226 the GH13 GBE from Escherichia coli (EcGBE13) was included as a reference

The claim of authors that Starch-like polymers can be created by the use of enzymatic modification with glycogen 10 branching enzymes (GBEs) is interesting and such a work should be carried on. Overall the quality of manuscript is nice. I recommend this manuscript for publication in polymers with following minor changes

i) English throughout the manuscript is average quality, the author should carefully check the grammatical/syntax errors.

ii) References cited by the authors in the results and discussion section should be updated, a lot of work in the same field is recently published and must be cited. I suggest the author should replace reference number 13,16,33,34 with new one

iii) The authors should rewrite the conclusion section and should make it more readers friendly

English through out the manuscript is of average quality, the author should carefully revise English throughout.

Author Response

We thank the reviewer for the comments and we appreciate the reviewers’ statement that the use of enzymatic modification with glycogen branching enzymes to create a starch-like polymer is interesting. We have read the point of improvement carefully and corrected the manuscript accordingly. See addressed points below.

1. English throughout the manuscript is average quality, the author should carefully check the grammatical/syntax errors.

The English in the manuscript is critically reviewed and we’ve adjusted the text. Changes are highlighted in lines 12-16; 70-73; 81-111; 227-234; 237-246; 253-272; 280-293; 299-309; 317-337; 365-378.

2. References cited by the authors in the results and discussion section should be updated, a lot of work in the same field is recently published and must be cited. I suggest the author should replace reference number 13,16,33,34 with new one

The reference number 16 has been replaced with a new reference (Moller, M.S., Henriksen, A., Svensson, B. (2016) Structure and function of α-glucan debranching enzymes. Cell. Mol. Life Sci. 73(14) 2619-41.), and the references 13, 33 and 34 were replaced with references already present in the reference list.

3. The authors should rewrite the conclusion section and should make it more readers friendly

The conclusion section has been critically reviewed and some changes have been made to the conclusion section to make it more easy to read.

Reviewer 2 Report

In this manuscript, the authors describe the "Alpha-1,4-transglycosylation activity of GH57 glycogen branching enzymes is higher in the absence of a flexible loop with a conserved tyrosine residue". While this manuscript is in general well written, it would be helpful if the authors address these minor concerns:

1) In the title itself, the highlighted word should be "are" instead of "is". It would be helpful if the authors revise this.

2) In lines 65-71, a schematic illustration describing the double displacement mechanism of GH57 GBE's would be very helpful.

3) In line 338, to avoid unnecessary distractions to potential readers, it will be helpful if figures and their descriptions are placed on the same page. A similar issue can be found in line 359 for figure 4. The authors should consider revising this.

4) In all the GH57 glycogen branching enzymes (GBE's) studied, was beta-1,4-transglycosylation activity ever detected? What about alpha-1,6-transglygosylation activity?

5) How was the stereochemistry of these enzyme-catalyzed transglycosylation reactions determined in this study?

In summary, this manuscript could potentially benefit its target readers if the above concerns are adequately addressed.

Author Response

We thank the reviewer for the comments and we appreciate the reviewers’ statement that the manuscript is well written. We have read the point of improvement carefully and corrected the manuscript accordingly. See addressed points below.

1. In the title itself, the highlighted word should be "are" instead of "is". It would be helpful if the authors revise this.

The verb "is" should be used instead of "are" to agree with the singular noun "activity." Since the subject "Alpha-1,4-transglycosylation activity" is singular, the verb should also be singular.

2. In lines 65-71, a schematic illustration describing the double displacement mechanism of GH57 GBE's would be very helpful.

The mechanism has been schematically illustrated before in other research. We added a line referring to these papers. “as schematically illustrated by Hayashi et al (2017) and Gaenssle et al (2021) [9, 28]” (line 66).

3. In line 338, to avoid unnecessary distractions to potential readers, it will be helpful if figures and their descriptions are placed on the same page. A similar issue can be found in line 359 for figure 4. The authors should consider revising this.

The order of figures and text is slightly adjusted, to ensure that all figures are on the same page as the first time they are described.

4. In all the GH57 glycogen branching enzymes (GBE's) studied, was beta-1,4-transglycosylation activity ever detected? What about alpha-1,6-transglygosylation activity?

In glycogen synthesis, only alpha-linkages are formed, no beta-linkages. Therefore, Beta-1,4-transglycosylation activity of GH57 GBEs has not been studied. Alpha-1,6-transglycosylation activity is present in GH57 GBEs, and in the manuscript often referred to as branching activity. This is highlighted in line 95.

5. How was the stereochemistry of these enzyme-catalyzed transglycosylation reactions determined in this study?

The stereochemistry was not determined in this study specifically, but it is well known from previous research that glycogen branching enzymes only catalyze alpha-linkages. Therefore, we can say that all linkages created by the enzymes in this study are also alpha-linkages.

Reviewer 3 Report

Overall, the work is well written, the experiments well planned and executed. The obtained data were correctly interpreted and allowed to formulate an answer to the question regarding the selected element of the structure of the tested enzymes. In my opinion, the work meets the requirements of the journal and deserves to be published in its current form.

Author Response

We thank the reviewer for carefully reading our manuscript. We appreciate the positive recommendation for publishing our manuscript in the current form.

No comments to address.